# 1946 and the Early History of Hydrosilylation

**DOI:** 10.3390/molecules27144341

**Published:** 2022-07-06

**Authors:** Kenrick M. Lewis, Sabine Couderc

**Affiliations:** 1Momentive Performance Materials Inc., 769 Old Sawmill River Rd, Tarrytown, NY 10591, USA; 2Momentive Performance Materials GmbH, Kaiser-Wilhelm Allee 1, 51373 Leverkusen, Germany; sabine.couderc@momentive.com

**Keywords:** hydrosilylation, history, Wagner, Speier, Sommer, platinum, peroxide, Ellis-Foster, Montclair, Union Carbide

## Abstract

Three events occurred in the second half of 1946 in three adjoining US States (NJ, NY, and PA) which marked the birth of Hydrosilylation Technology. They occurred before the landmark 1957 JACS paper and the 1958 issued US patent by Speier et al. and before Chalk and Harrod named the reaction. First, on 27 June 1946, Mackenzie et al., of Montclair Research Corp., applied for a patent to prepare addition compounds of hydridosilanes and unsaturated organic compounds. Then, on 9 October 1946, Wagner and Strother of Union Carbide Corp. applied for a patent on a process to produce organic compounds of silicon with Si–C bonds by reacting a hydridosilane and an alkene or alkyne in the presence of a catalyst metal of the platinum group. Finally, Sommer et al., submitted a paper on peroxide-catalyzed hydrosilylation to JACS on 17 December 1946. It was published in January 1947. The landmark patent interference ^§^ and priority ^§^ case law associated with the Mackenzie et al. and Wagner et al., applications is well known to patent attorneys. This presentation will retrace the origins of hydrosilylation and report events (1946–1960) in the history of the reaction that are most probably unknown to most authors and presenters of hydrosilylation investigations. George Wagner’s contribution to the birth of this technology is also highlighted.

## 1. Introduction

Authors of papers on hydrosilylation, as well as presenters on hydrosilylation at conferences, customarily acknowledge the contribution of John Speier to platinum-catalyzed hydrosilylation. The certainty of the homage and attribution of first inventorship is further emphasized by the named reagent, Speier’s Catalyst (H_2_PtCl_6_ in isopropanol), and the combined greater than one thousand citations of the seminal JACS paper [1] and US patent [2]. Unrecognized is the fact that, in their February 1957 JACS paper [1] and February 1958 issued patent [2], Speier et al. cited three prior art ^§^ patents [3,4,5] which disclosed the use of platinum and other Group VIII metals and salts as catalysts for the addition of SiH functional groups to alkenes and alkynes. Less well known are the pioneering inventions of George Wagner and Corneille Strother [4,5]; the ground-breaking patent application of Mackenzie, Spialter, and Schoffman [3]; and the landmark patent interference ^§^ and priority ^§^ case law [6] associated with hydrosilylation. Herein, we retrace the origins of hydrosilylation and report events (1946–1960) in the history of the reaction that are most probably unknown to most authors and presenters of hydrosilylation investigations.

## 2. Hydrosilylation Inventions in 1946

The desire and need to produce organofunctional silanes and siloxanes via silicon-carbon bond formation had been evident in the immediate post-World War II era [7,8,9,10,11]. Rochow’s Direct Process [12,13] was practical only for methylchlorosilanes and phenylchlorosilanes and not for other alkyl or aryl analogs. Alternative synthetic methods were laborious and unattractive for commercial scale manufacture. Most often, they focused on Grignard and similar transformation of Si–Cl bonds to Si–Alkyl and not on Si–H to Si– Alkyl as contemporary review articles by Burkhard et al. [10] and George et al. [11] reveal. Independently, three groups, all located in the adjoining States of New Jersey, New York and Pennsylvania, pursued the addition of the Si-H functionality to unsaturated carbon-carbon bonds and their efforts [3,4,5,9,14] came to successful fruition in 1946. So, 1946 can be considered the Hydrosilylation Year (See Table 1), although the term, hydrosilylation (also called hydrosilation), was coined by then GE scientists, Allan Chalk and John Harrod, in 1965 [15,16]. They also proposed the well-known Chalk–Harrod hydrosilylation mechanism [15,16,17]. Curiously, however, the early patents (Table 1, Entries 1–6) highlighted here are not cited in any of their publications.

On 27 June 1946, Mackenzie, Spialter, and Schoffman [3] applied for a US Patent to prepare addition compounds of hydridosilanes and unsaturated organic compounds. Montclair Research Corp. was the assignee ^§^. This application (US Appl. 679,856) was abandoned some time later and replaced with a continuation-in-part ^§^, which was issued as US 2,721,873 on 25 October 1955. Consequently, there is no issued patent associated with the original application number. However, a French Patent application [3] (see Table 1, Entry 2) was made on 21 February 1948. The grant was made on 28 November 1949 and the patent (FR 961,878) was published on 24 May 1950. It is from this document and from US 2,721,873 [19] that we learn the details of the Mackenzie et al. invention. *Note that Speier et al. mistakenly cite the French Patent as 961,876 and that this error might have contributed to the ignorance of this patent by hydrosilylation researchers*.

Mackenzie et al. [3,19] disclose and claim ^§^ thermal reaction of hydridosilanes with alkenes and alkynes under autogenous conditions to form addition compounds that are alkylsilanes or alkenylsilanes. The illustrative examples ^§^ cover synthesis of C1–C12 alkyltrichlorosilanes as well as cyclohexyltrichlorosilane and phenethyltrichlorosilane. One experiment was conducted with UV radiation. However, there was no claim on the use of light, metallic catalysts or peroxides, despite their disclosure in the specifications (see Columns 5 and 6 of US 2,721,873).

On 9 October 1946, Wagner and Strother [4] of Union Carbide Corp. applied for a patent (Table 1, Entry 3) on a process to produce organic compounds of silicon with Si-C bonds by reacting a hydridosilane and an alkene or alkyne in the presence of a catalyst metal of the platinum group. Platinum black and platinized asbestos were the preferred catalysts. Patent application was also made in Switzerland and a Swiss patent, CH-279,280 [5], was granted on 30 November 1951 and published on 1 March 1952. The US Patent (US 2,632,013) was issued on 17 March 1953, more than six years after the original application date. Improvements of the hydrosilylation process were disclosed in later patents [20,21] (Table 1, Entries 4 & 6) claiming use of platinum on charcoal (US 2,637,738) [20] and platinum on gamma-alumina (US 2,851,473) [21] as catalysts.

Professor Leo Sommer and co-workers submitted a paper [14] on peroxide-catalyzed hydrosilylation to JACS on 17 December 1946. This paper was published one month later in January 1947. It was the first public disclosure of a catalyzed hydrosilylation. However, note that Mackenzie et al., in US 2,712,873, had included peroxides among the catalysts they specified (but did not claim) for the addition of hydridosilanes to alkenes and alkynes. Peroxides had also been listed as catalysts in two patents [22,23] assigned to E. I. du Pont de Nemours & Co. These patents disclosed high temperature (>450 °C) autogenous processes to synthesize organosilicon compounds via reaction of chlorosilanes with aromatic and aliphatic hydrocarbons.

Kanner’s *History of Union Carbide Silicones* [24] contains a memorandum, dated 24 December 1945, co-authored by Strother, Wagner, and McConnell summarizing the status of their research in organosilicon chemistry and proposing catalytic addition of Si–H to alkenes and alkynes to obtain alkyl and alkenyl halosilanes. It is reproduced here to showcase its prescience and insight. Note also the speculation about hydroboration, six years before its experimental realization.

### 2.1. The Christmas Eve 1945 Memorandum

#### 2.1.1. Object

The purpose of this memorandum is to indicate the scope of the field of organo-silicon chemistry, to evaluate the active patent art in this field, and to summarize our present research position.

#### 2.1.2. Summary

Organo-silicon chemistry appears to be potentially a field as broad as that of carbon chemistry. The active patent art covers only a small sector of the potential field. There seems to be enough old literature to prevent any one company from monopolizing organo-silicon chemistry, even the small section on silicone resins and oils which is receiving so much publicity today. The Laboratory has developed several interesting reactions which are believed to give us an entrée into the field.

#### 2.1.3. Current Program

Work at this Laboratory has developed a reaction for preparing the organo-silicon halides. The method is believed to be more flexible than the General Electric reaction between hydrocarbon halides and silicon since the latter reaction requires an organic radical which is stable at high temperatures.

Our process may be represented in the steps shown below:HCl + Si → SiHCl_3_; SiH_2_Cl_2_; etc. A copper catalyst is used in the reaction.(a)SiHCl_3_ + olefins → RSiCl_3_(b)SiH_2_Cl_2_ + olefins → R_2_SiCl_2_

The first step, including use of a copper catalyst, was discovered by Combes (Compt. Rend. 127, 531 (1896) and 148, 555 (1909)) and is, therefore, not broadly patentable. Our second step is carried out at 200–250 °C and is facilitated by application of pressures of the magnitude employed in producing polyethylene. This second step is not shown in the prior art although duPont in US Patent 2,379,821, issued 3 July 1945, claims the reaction between aliphatic hydrocarbons and inorganic silicon halides having at least one halogen atom of atomic weight above twenty at temperatures of at least 450 °C. Aromatic hydrocarbons are disclosed in the specification of this patent. The temperature qualification is a part of all the granted claims which therefore do not read on our process. It might be pointed out that the duPont patent is concerned with the reaction of any hydrocarbon with a halogen on a silicon atom while our step 2 is concerned with the reaction of any unsaturated organic compound with a hydrogen on a silicon atom.

It is anticipated that this reaction may be applicable to any compound containing a double bond. Should this be the case, a general method would be available for introducing various organic groups into the siloxanes. To illustrate, a siloxanyl radical on a ketone or cyanide group might produce a plasticizer with flame resistant properties.

In addition to the siloxanes, the reactions of silicon hydrides with carbon double bonds may permit the formation of multiple silicon-carbon chains or the substitution of silicon for carbon in any organic compounds. For example, the reaction of silicon hydrides with butadiene may give a five member ring containing silicon or, with acetylene, may give vinyl silicon hydrides capable of extensive reaction. Furthermore, it is possible that the addition of a hydride to a double bond may not be confined to the silicon hydrides. In this case organic compounds containing sodium, boron, phosphous, arsenic and the like could be prepared from their respective hydrides, NaH, B_2_H_6_, PH_3_, or AsH_3,_ etc. In manfacturing these, the raw materials would be avialable within the Corporation since the hydrides for the most part are made from products of the electric furnace. It is contemplated not only that such products will necessarily find markets for themselves but also that they will serve as intermediates in chemical industry.

#### 2.1.4. Conclusions

Only a small percentage of the possible derivatives of the siloxanes have been prepared and only a few of these are patented. While the validity of certain of these patents appears to be questionable, it should be understood that the suggestions herein on anticipation by prior art are of a tentative nature for consideration and final decision by the Patent Department. It is recommended that their opinion on this matter be sought.

The work in this Laboratory has established a background in a field of considerable breadth. The work has already led to reactions of importance.

The experience of several years has bred a comprehension of the extent of the industrial possibilities of organo-silicon compounds as well as other related organometa1lic compounds. It is felt that the business outlook will justify the utilization in additional research of the knowledge and viewpoint now acquired.


C.O. Strother, G.H. Wagner and C.W. McConnell.


The memorandum disclosed that thermal hydrosilylation of olefins with trichlorosilane and dichlorosilane had already been achieved at 200–250 °C and “pressures of the magnitude employed in producing polyethylene”. Union Carbide exhibits presented in the hearing [6] before the Appeals Court confirmed that this *thermal hydrosilylation* invention occurred in August 1945. Wagner’s memoir [9] gives the date of the *platinum catalyzed invention* as “early 1946”.

Dallas Hurd [25] of General Electric Co. disclosed the thermal hydrosilylation of olefins by SiH_4_ in US 2,537,763 (9 January 1951). The application was made less than a year earlier on 8 February 1950. Alkylated silanes of general formula, R_n_SiH_(4−n)_, (*n* = 1–4, R = ethyl, isobutyl), were synthesized in the illustrative examples included therein.

## 3. Aftermath of the Publication of the Hydrosilylation Patents

Following the publication of US 2,632,013 (the first Wagner and Strother patent), in March 1953, Montclair Research Corp (by then merged with Ellis-Foster Company) amended its abandoned application to include claim 2 of the patent and the Primary Examiner ^§^ declared an interference [6]. Claim 2 reads on thermal, uncatalyzed hydrosilylation, not platinum-catalyzed hydrosilylation. The interference was sent to The Board of Patent Interferences for a final hearing and decision. Ellis-Foster/Montclair chose not to offer testimony. Union Carbide “took testimony and introduced exhibits to show that the invention had been conceived about August 1945 and had been reduced to practice during the period from November 1945 to February 1946.” [6]. This is important because, at the time, priority of the invention in the USA was based on which party first had the idea (duly signed, dated, and witnessed and diligently investigated). Since 2011, priority is based on first to file. The Board decided that “ since the plaintiffs (Ellis-Foster/Montclair) could not make a valid claim for the invention because their specifications did not disclose it, priority had to be awarded to the defendant (Union Carbide), there being no valid claim in the plaintiffs on which any assertion of priority could be bottomed by them” [6]. However, Ellis-Foster/Montclair appealed first to the United States District Court, which upheld the Board’s decision, and later to United States Court of Appeals Third Circuit. Arguments before the latter were heard on 10 June 1960. Its decision [6] reversing the lower court order was handed down on 3 November 1960. So, Ellis-Foster/Monclair’s claim to thermal hydrosilylation was upheld, but Union Carbide’s claims to Pt-catalyzed hydrosilylation were unaffected.

Alas, there was yet more legal activity. Union Carbide filed a writ of certiorari with the US Supreme Court requesting a review of the judgement of the Third Circuit Court. A writ of certiorari orders a lower court to deliver its record in a case so that the higher court may review it. The Supreme Court denied the request [26], thereby letting the Third Circuit decision stand.

## 4. Commercialization of Hydrosilylation-Based Products

In the meantime, Union Carbide reduced US 2,632,013 and US 2,851,473 to commercial practice [8,9,27,28,29,30,31,32] and was already producing vinyltrichlorosilane (and vinylalkoxysilanes) at the Tonawanda, NY, semiworks plant in 1949 and the Sistersville, WV, manufacturing plant in 1955, by the time Speier et al. applied for their patent. Any available unsaturated material was subjected to hydrosilylation; only acrylonitrile proved resistant [9]. The various chlorosilanes and alkoxysilanes made were new to the world at that time. They were sampled to academic [28,29,30] and industrial institutions [8,9,31,32,33,34,35,36,37,38] around the world for investigation of their properties and possible uses. Since Union Carbide was a major producer of ethylene, acetylene and other unsaturated gaseous hydrocarbons [39], it was natural for Wagner to use those as raw materials in his research and development work. Additionally, the group at Tonawanda was skilled in the use of pressurized reactors and autogenous processes [8,24]. So, heterogeneous hydrosilylation catalysis over supported platinum was the method chosen and the heterogeneously-catalyzed hydrosilylation of acetylene by trichlorosilane was practiced commercially at the Sistersville, WV, Plant well into the 1980’s. It is noteworthy that H_2_PtCl_6_ was the source [2,20] of Pt for both the Wagner et al. (*heterogeneous catalysis*) and Speier et al. (*homogeneous catalysis*) inventions.

Kanner [24] and Stief [39] both attribute Union Carbide’s successful penetration in the silicones and silanes markets to its hydrosilylation technology. Although Dow-Corning and General Electric had had almost a decade head start, the uniqueness of hydrosilylation enabled Union Carbide to produce specialty products that customers wanted that the other two could not. A massive effort in R&D, manufacturing, marketing, sales and technical service for organofunctional silanes and silicone surfactants underpinned that success.

The Direct Synthesis of ethylchlorosilanes from ethyl chloride and copper-activated silicon is selective for ethyldichlorosilane and ethyltrichlorosilane, not diethyldichlorosilane [7]. So, over a ten-year period (1945–1955), Union Carbide devoted considerable R&D to the synthesis, characterization and applications of diethylsilicones derived from diethyldichlorosilane made via Pt-catalyzed hydrosilylation of dichlorosilane with ethylene. Diethylsilicones were viewed as competitive replacements for dimethylsilicones derived from Direct Process dimethyldichlorosilane. However, the differences in chemical and physical properties, especially oxidative stability and the tendency to form cyclics, between poly(dimethylsiloxanes) and poly(diethylsiloxanes) were too great to enable drop-in replacement [24]. Union Carbide therefore licensed General Electric’s Direct Process to get access to poly(dimethylsiloxanes) [24,39]. Manufacture of methylchlorosilane monomers began at the Sistersville, WV, Plant in 1955 [32].

## 5. Conclusions

George Wagner’s experimental results fulfilled the objectives set forth in the 1945 Christmas Eve memorandum. The availability of platinum-catalyzed hydrosilylation enabled the synthesis and manufacture of a large variety of organofunctional silanes and siloxanes [7,8,9,33,34,37,38], which have found use in glass fiber composites [8,37,38] heat resistant elastomers [37,38], polyurethane foams [8,24,27], and many other materials. The impact of this hydrosilylation-enabled molecule to material innovation endures to the present time. Academic and industrial practitioners of hydrosilylation are aware that products made using platinum-catalyzed hydrosilylation contribute to the satisfaction of basic human needs like food, clothing, shelter, healthcare, transportation, and communication. However, George Wagner’s contribution to hydrosilylation technology remains largely unrecognized. It is hoped that this review will not only teach about the legal history of hydrosilylation technology, but will also inform about Dr. Wagner’s pioneering influence on it.

## 6. Tribute to George Wagner

Dr. Wagner got his PhD in physical chemistry from the University of Iowa in 1941 and began employment in the Linde Division of Union Carbide in Tonawanda, NY, in the same year. His initial research assignment was to develop thermally stable synthetic lubricants for military aircraft. He did so successfully with compositions comprising aromatic amine antioxidants and polypropylene oxide polymers. These synthetic lubricants were used in fighter aircraft during the Pacific campaign in World War II [24]. His memoir [9] recounts how the first platinum-catalyzed addition of unsaturated hydrocarbons and hydridosilanes came about. Dr. Wagner was inventor of forty-one patents and author of thirteen scholarly publications before leaving the technical career path for the management ladder in 1959. Notable among his non-hydrosilylation inventions are increased H_2_SiCl_2_ formation with the use of copper to catalyze the Direct Reaction of HCl and Si [40]; SiH_4_ synthesis via catalytic disproportionation of HSi(OR)_3_ (R = CH_3_, C_2_H_5_) [41]; various methods for synthesis of phenylsilanes [42,43,44], alkoxysilanes [45], and chlorosilanes [46,47]; allyltrichlorosilane synthesis via aluminum chloride or quinoline-catalyzed dehydrochlorination of 3-chloropropyltrichlorosilane [48,49]; and polymerization of ethylcyclosiloxanes under high pressure at room temperature [50,51,52]. Availability of H_2_SiCl_2_ was crucial to Union Carbide’s R&D on diethyldichlorosilane and poly(diethylsiloxanes). Dr. Wagner’s research [9,24,41,46,47] on the preparation of high purity silicon from SiH_4_ and HSiCl_3_ provided the basis for future research [53] on this subject within Union Carbide. Dr. Wagner was Research Director of the Linde Division of Union Carbide from 1959–1969 and Vice President of the Metals and Mining Division from 1969 until his retirement in 1971. In 1960, he was awarded the Schoellkopf Medal [8,54] by the Western New York Section of the American Chemical Society for “work with propylene oxide polymers and other stabilizers used as superior lubricants”. Post-retirement, he was Adjunct Professor of Geology, 1974–1999, at the University of Arkansas where he was active in research and published in scholarly journals. His enduring legacy is the platinum-catalyzed addition of hydridosilanes to unsaturated compounds. He passed away in March 2004, aged 89.

## Figures and Tables

**Table 1 molecules-27-04341-t001:** Key patents ^§^ and publications in the history of hydrosilylation.

Entry	Filing Date & Application Number	Inventors & Assignees	Key Claim	Date Issued & Comments
1	27 June 1946, US Appl# 679,856	Mackenzie, Spialter & Schoffman to Montclair Research Corp	Heating SiH and unsaturated organic compounds to form addition compound	Original US application abandoned. Continuation in part granted as US 2,721,873 (25 October 1955)
2	French Application, 21 February 1948	Ditto	Ditto	French Patent FR 961,878, * 24 May 1950
3	9 October 1946 US Appl # 702,084 7 October 1947 Swiss Application	Wagner & Strother to Union Carbide Corp. Linde Air Products (A Division of Union Carbide)	Process to produce silicon compounds with Si-C bonds by reacting a hydridosilane with an alkene or alkyne in the presence of a platinum metal catalyst	US 2,632,013 (17 March 1953). Swiss Patent CH-279,280 (30 November 1951), published (1 March 1952)
4	17 September 1949	Wagner to Union Carbide Corp	Pt on charcoal as catalyst	US 2,637,738. 5 May 1953
5	8 February 1950	Hurd to General Electric Co.	Method of making alkylsilanes by heating olefins and SiH_4_	US 2,537,763 9 January 1951
6	23 December 1955	Wagner & Whitehead to Union Carbide Corp	Pt on gamma Al_2_O_3_ as catalyst	US 2,851,473 9 September 1958
7	5 December 1955	Speier & Hook to Dow Corning	H_2_PtCl_6_ as catalyst	US 2,823,218 11 February 1958
**Scholarly Articles**	**Date Received**	**Authors**	**Key Disclosure**	**Reference**
8	17 December 1946	L.H. Sommer, et al.	Peroxide catalyzed hydrosilylation	JACS, **1947**, *69*, 188 [14]
9	17 October 1955	J.L. Speier, et al.	Peroxide catalyzed hydrosilylation	JACS, **1956**, *78*, 2278 [18]
10	20 August 1956	J.L. Speier, et al.	H_2_PtCl_6_ catalyst for hydrosilylation	JACS **1957**, *79*, 974 [1]

* Cited by Speier et al. [1] with an incorrect French Patent number, ^§^ A glossary of terms commonly used in patents is provided at the end of this article.

## Data Availability

This study did not report any data.

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
