# Peer review of "1946 and the Early History of Hydrosilylation"

_molecules, 2022, doi:10.3390/molecules27144341_

Round 1

Reviewer 1 Report

Manuscript “1946 And The Early History Of Hydrosilylation” by Lewis and co-worker is a fascinating review on the unknown pioneering work and legal actions involved in the early history of the hydrosilylation process. The manuscript, clearly written and of enjoyable reading, will be of interest for many readers of Molecules and not only for those familiar with silicon chemistry.

I recommend publication after corrections of minor spelling/editing mistakes listed below.

-) line 82: the word “process” is written in different size

-) it is not clear the exact beginning and ending of the citation/extract from Kanner’s memorandum; can the authors better highlight this part?

-) line 195: change “hydosilylation” into “hydrosilylation”

-) line 240: change “fulifilled” into “fulfilled”

-) line 266: change “qunoline-catalyzed” into “quinoline-catalyzed”

Author Response

The typographical errors in lines 82, 195, 240 and 266 have been corrected as indicated by the tracker

Subsection 2.1 The Christmas Eve 1945 Memorandum has been added to demark the start of the excerpt from Dr. Kanner’s History of Union Carbide Silicones.  A signature line with the names of Strother, Wagner and McConnell marks the end

Reviewer 2 Report

The Authors described the early history of hydrosilylation with a particular emphasis on the priority of the publication. I really enjoyed reading this document. The analysis of very old patents and papers revealed facts that can be unknown even to a broader audience in the field of hydrosilylation/organosilicon chemistry. I think that review articles like this one are really important and deserve to be published because they remind us about the history of important chemical processes that influenced the world. I recommend this manuscript for publication. I think that a manuscript can be published as it is.

Author Response

My co-author and I thank Reviewer #2 for the favorable comments and review

Reviewer 3 Report

The review of 1946 And The Early History Of Hydrosilylation makes a good impression. I really enjoyed reading it. Learned a lot of interesting things for myself. I did not know about many things about hydrosilylation before reading the review. I think that this review will be useful for many chemists involved in organosilicon chemistry. The only remark: the problem of direct synthesis of diethyldichlorosilane was solved in the USSR.

I recommend this review for publication.

Author Response

Thanks to Reviewer # 3 for the favorable comments and recommendation that the manuscript be published